

# Deriving customized terrain classes for avalanche risk management in mechanized skiing operations from operational terrain assessments

Reto Sterchi, Pascal Haegeli

School of Resource and Environmental Management, Simon Fraser University, Burnaby, BC, V5A 1S6, Canada

*Correspondence to*: Reto Sterchi (reto_sterchi@sfu.ca)

**Abstract.** An in-depth understanding of the nature of the available terrain and its exposure to avalanche hazard is crucial for making informed risk management decisions when travelling in the backcountry. While the Avalanche Terrain Exposure Scale (ATES) is broadly used for providing recreationists with terrain information, this type of terrain classification has so far only seen limited adoption within the professional ski guiding community. We hypothesize that it is the generic nature and small

number of terrain classes of ATES and its precursor systems that prevent them from offering professional decision makers meaningful assistance. Working with two mechanized skiing operations in British Columbia, Canada, we present a new approach for deriving terrain classifications from daily terrain assessment records. We used a combination of self-organizing maps and hierarchical clustering to identify groups of ski runs that have been assessed similarly in the past and organized them into operation-specific terrain hierarchies. We then examined the nature of the emerging terrain hierarchies using

comprehensive run characterizations from experienced guides. Our approach produces high-resolution terrain hierarchies that offer a more nuanced and meaningful perspective on the available skiing terrain and provide new opportunities for examining professional avalanche risk management practices and developing meaningful decision aids.

## 1   Introduction

Commercial mechanized backcountry skiing is a type of downhill skiing where guided groups use helicopters or snowcats to

access remote and pristine skiing terrain that would otherwise be difficult to access. In Canada, the birth place of mechanized skiing, this sector is a substantial part of the local skiing industry, providing more than 100,000 skier days per winter (HeliCat Canada, 2016). Since its inception in the late 1960s, the Canadian mechanized skiing industry has provided roughly 3 million skier days in total (HeliCat Canada, personal communication; Walcher et al., under review). While most of the global mechanized skiing activity is taking place in Canada, it is also offered in other parts of the world including the United States,

Iceland, Greenland, South America and the Caucasus region.

Skiing untracked powder in uncontrolled mountain terrain is not without risk. Skiers are exposed to numerous types of natural hazards that can lead to injury or even death. Snow avalanches are the greatest natural hazard affecting the mechanized skiing industry in Canada (Bruns, 1996). Walcher et al. (under review) documented that between 1970 and 2016, the Canadian mechanized skiing industry experienced a total of 81 avalanche fatalities in 44 accidents involving both guides and guests.





During the last two decades (1997-2016), the risk of accidentally dying in an avalanche was calculated as 14.4 micromorts (number of deaths per million skier days), which represents 77% of the overall mortality in mechanized skiing in Canada due to natural hazards during that period (Walcher et al., under review).

While the risk from avalanches can never be eliminated completely, mechanized skiing operations aim to provide their guests

with a high-quality skiing experience without exposing them to an unacceptable level of risk (McClung, 2002; Israelson, 2015). The primary strategy for managing the risk from avalanches when travelling in the backcountry during the winter time is to limit one's exposure by carefully choosing when and where to travel (Statham, 2008; Canadian Avalanche Association, 2016). Thus, identifying terrain that is appropriate under different types of avalanche conditions is crucial for making informed decisions when travelling in the backcountry.

Mechanized skiing operations in Canada follow a well-established process for selecting skiing terrain. This process is iterative in nature and has been described as a series of filters occurring at multiple spatial and temporal scales (Israelson, 2013, 2015). Each morning, guiding teams go through their inventory of predefined ski runs and collectively decide which runs are open or closed for skiing with guests under the expected avalanche hazard conditions. These predefined ski runs may be as large as a small ski resort, and there may be multiple distinct ways of skiing them from the available helicopter landings at the top of the

run to the pickup locations at the bottom. The resulting large-scale, consensus-based "run list" has established itself as a critical component in the risk management process of many commercial backcountry skiing operations (Israelson, 2013) and is considered best practice within the industry. The run list represents the foundation for all subsequent guiding decisions during the daily skiing program by explicitly specifying the set of ski runs that have acceptable ski lines under the current conditions. Over the course of a skiing day, terrain choices are further refined and adapted using real-time field observations. While

avalanche hazard is one of the most critical factors in this process, other factors such as weather and flying conditions, flight economics, skiing quality, guest preferences and skiing abilities also affect the selection and sequencing of skied terrain (Israelson, 2015).

Bruns (1996) and later Adams (2005) have described that senior guides make their risk management decisions to a considerable part intuitively, using experience-based heuristics without necessarily reviewing every aspect of the decision situation

conscientiously. While research in cognitive psychology has shown that experience-based heuristics can perform well under uncertainty (e.g., Gigerenzer and Gaissmaier, 2011), they can also lead to erroneous outcomes if not applied appropriately (e.g., McCammon, 2002). Despite the well-established, systematic approach to terrain selection, the misapplication of terrain remains among the most common errors of professional guides in the mechanized skiing industry (Guyn, 2016). To assist guides in their daily terrain selection, there have been various attempts to classify the severity of ski runs. Canadian Mountain

Holidays (CMH), a large mechanized skiing provider that operates twelve lodges in the Columbia Mountains of western Canada, developed an ordinal severity rating system for their ski runs in the late 1980s (J. R. Bezzola: personal communication). Based on the expert opinion of long-time guides working at each lodge, this system assigned all ski runs into one of three increasingly severe terrain classes ranging from *Class A* (forgiving terrain that needed little investigation and could be skied safely in most conditions) to *Class B* (terrain that is moderately difficult to assess considering historical climatic





conditions and that has moderate consequences in case of a mishap) and *Class C* (complex terrain with severe consequences in case of a mishap and which needed more extensive investigation before being skied) (Canadian Mountain Holidays, 1994). Despite considerable efforts by CMH, the terrain classification system did not establish itself as an operational tool for making run lists. Experienced guides did not find that the rating system added value as they perceived the classes to be too general and

the system too restrictive for meaningful decision-making (J. R. Bezzola: personal communication). The three-class rating system was eventually abolished in the mid-1990s.

To provide amateur recreationists with a tangible tool for making terrain choices when planning a backcountry trip, Statham et al. (2006) developed the Avalanche Terrain Exposure Scale (ATES). Like the original system of CMH, the objective of ATES was to provide users with an overall severity assessment of linear backcountry trips into avalanche terrain that is easy

to understand and communicate. The system considers eleven terrain parameters (e.g., slope angle, slope shape, terrain traps, route options, etc.) and classifies trips into three ordinal classes. *Simple* terrain is characterized by exposure to low angle or primarily forested terrain. Some forest openings may involve the runout zones of infrequent avalanches but many options to reduce or eliminate exposure may exist. *Challenging* terrain is described as being exposed to well defined avalanche paths, start zones or terrain traps. Options to reduce or eliminate exposure exist, but require careful route finding. *Complex* terrain,

the most severe class, is characterized by multiple overlapping avalanche paths or large expanses of steep, open terrain with multiple avalanche start zones and terrain traps below with minimal options to reduce exposure (Statham et al., 2006). Since the initial introduction of ATES, many backcountry trips in Canada have been rated according to the system (e.g., https://www.pc.gc.ca/en/pn-np/mtn/securiteenmontagne-mountainsafety/avalanche/echelle-ratings). And at the time of this writing, Avalanche Canada has mapped more than 8,000 km$^2$ of avalanche terrain in western Canada using the ATES mapping

approach developed by Campbell and Marshall (2010), Campbell et al. (2012) and Campbell and Gould (2013)(K. Klassen: personal communication). Today, ATES ratings are a critical component of the Canadian avalanche awareness curiculum and public avalanche safety products, such as the trip planning tool of the Avaluator V2.0 decision aid (Haegeli, 2010a) and its online implementation (https://www.avalanche.ca/planning/trip-planner). The system has also been adopted in other countries including Spain (Gavaldà et al., 2013; Martí et al., 2013), Sweden (Mårtensson et al., 2013) and Switzerland (Pielmeier et al.,

25   2014).

Even though it has been hypothesized that many guides conceptualize the ski runs of their operation in groups with a hierarchical structure (J. R. Bezzola: personal communication), the response of the mechanized skiing community to the ATES system has so far been limited. Northern Escape Heli-Skiing (NEH) initially tried to use the ATES system for classifying their ski runs but found it to be far too conservative for professional use in commercial heli-skiing (Israelson, 2013). Consequently,

NEH developed its own qualitative avalanche terrain severity rating system, which classifies individual ski lines according to their overall exposure to avalanche hazard on a three-class scale (Israelson, 2013).

There is no doubt that terrain classifications can play an important role in backcountry avalanche risk management. The broad use of ATES among amateur recreationists and the repeated attempts to introduce similar systems in mechanized skiing operations clearly highlight their potential value. But why have these efforts only had limited success in mechanized skiing



operations so far? We believe that the generic definitions and the small number of classes (i.e., limited resolution) of the existing systems are unable to characterize ski runs in a way that can offer meaningful insight to professional guides for their risk management decisions beyond just showing the obvious. But how can a more useful terrain classification system be created for mechanized skiing operations?

There has been considerable research that aims to better understand the link between terrain and avalanche hazard. Most of it has taken a natural science perspective to relate patterns of well documented avalanche occurrences to geomorphologic parameters. This approach has linked relatively easily accessible geomorphologic parameters, such as incline or curvature, with the frequency or likelihood of avalanches (Schaerer, 1977; Smith and McClung, 1997; Maggioni and Gruber, 2003). Moreover, automated procedures based on digital elevation models have been developed to identify potential avalanche release

areas as input for numerical avalanche runout modeling (Maggioni and Gruber, 2003; Bühler et al., 2013) or mapping avalanche terrain (Delparte, 2007). While this area of research provides valuable input for land-use planning and the protection of permanent structures, it has so far only offered limited tools for backcountry risk management. Grimsdottir (2004) used questionnaires and interviews to examine the terrain selection process of professional guides. While her research highlighted individual terrain characteristics that influence the decision process of guides (e.g., terrain shape, slope size), it did not produce

a tangible tool for assessing the overall severity of ski runs and for deriving terrain classes.

The objective of our study is to introduce an alternative method for deriving terrain classes that offer more meaningful insight into terrain decisions in commercial mechanized skiing operations. Instead of building the classification on physical terrain characteristics, we derive the terrain classes from patterns in revealed terrain preferences reflected in past daily run list ratings. Our assumption is that ski runs that are considered open and closed for guiding under similar conditions will represent more

meaningful groupings for operational decision-making. We hypothesize that each operation has a unique, finely differentiated hierarchy within its ski runs that emerges from the available skiing terrain, the local snow climate and the particular skiing product this operation offers to their clients. Furthermore, we suspect that the details of the run hierarchies might differ from year to year in response to the particular conditions of the individual winters. We will use historic run list data from two commercial mechanized skiing operations to illustrate our approach and explore these research questions in detail.

The remainder of the paper is organized as follows: Section 2 introduces the study sites, offers an overview of the dataset and describes our two-step approach for identifying groups of ski runs and combining them into a run hierarchy. In section 3, we present the identified hierarchies of ski runs and describe the nature of the identified groups. We conclude by discussing the implications of our results for terrain management and professional decision-making in mechanized skiing.

## 2    Data and methods

Our method for developing a useful ski run classification for mechanized skiing operations applies a modern clustering approach to multi-season records of daily run list ratings that combines the advantages of an unsupervised machine learning algorithm with traditional hierarchical clustering. To better understand and describe the nature of the emerging hierarchy of



ski run groups, we had a senior lead guide in each participating operation independently provide comprehensive characterizations of all the runs included in our study. Since guides' terrain choices are driven by more factors than just the hazard potential, our run characterization included a wide range of relevant attributes. In our final step of the analysis, we applied hierarchical clustering to the typical run list rating time series of the identified run groups for each season individually to examine how the nature of specific winters can affect the run classification. The following sections describe the various components of our analysis in more detail.

## 2.1 Study sites

We used data from two commercial helicopter-skiing companies—Northern Escape Heli-Skiing and Canadian Mountain Holidays Galena—that operate in different types of skiing terrain, snow climates, and offer skiing products with a distinct focus. *Northern Escape Heli-Skiing* (NEH) is located in Terrace, British Columbia, and their operating area in the Skeena Mountains spans an area of nearly 6,000 km$^2$. NEH has been operating for 14 years, typically running a skiing program with multiple helicopters serving either single or multiple small groups. The elevation of the available skiing terrain ranges from 500 m to 2,000 m above sea level. While their entire tenure has 260 established ski runs, much of their skiing is focused on approximately 80 ski runs in their home drainage called Promised Land. Our study will focus exclusively on the ski runs located in Promised Land, which range in size between 0.1 km$^2$ and 2.8 km$^2$. The character of the local snow climate is maritime with storm slab avalanche problems during or immediately following storms being the primary avalanche hazard concerns and warm temperatures promoting rapid stabilization (McClung and Schaerer, 2006; Shandro and Haegeli, 2018).

*Canadian Mountain Holidays Galena* (CMHGL) is based out of a remote lodge in the Selkirk Mountains near Trout Lake, British Columbia, roughly 75 km southeast of Revelstoke. Their tenure area consists of approximately 1,200 km$^2$ of skiing terrain ranging from 850 m to 2,850 m above sea level and includes 295 established ski runs, which range in size between 0.1 km$^2$ and 19.1 km$^2$. CMHGL has been operating for 28 years, typically running a skiing program with a single helicopter that serves three or four groups of 11 skiers each. The tenure area of CMHGL is located in a transitional snow climate with a strong maritime influence (Haegeli and McClung, 2003). The two most important types of persistent weak layers in this area are crust-facet combinations due to rain-on-snow events in early season and surface hoar layers during the main winter months (Haegeli and McClung, 2003). Thus, avalanche hazard conditions with a combination of storm and persistent slab avalanche problems types are frequent (Shandro and Haegeli, 2018).

## 2.2 Identifying run groups and overall terrain hierarchy

While NEH and CMHGL both have extensive operational databases that include field observations, hazard assessments and records of terrain choices, the primary data used in this study are daily run list ratings that describe the suitability of the ski runs for guiding guests under the existing hazard conditions. In both operations, the guiding team codes runs or ski lines as either "Open for guiding", "Closed for guiding" or "Not discussed" every morning of the season. In addition to these standard





codes, CMHGL also uses "Conditionally open for guiding" (i.e., can only be considered for skiing if a specified condition is fulfilled which has to be determined in the field) and NEH uses "Closed for guiding for other reasons than avalanche hazard" (e.g. crevasses, open creeks, ski quality). While CMHGL does not have an explicit code for identifying runs that are closed for other reasons than avalanche hazard, it is common practice at this operation that these types of runs would not be discussed.

The complete dataset for CMHGL consists of 469,280 run list ratings for 295 ski runs from 2,029 days during 18 winter seasons between 2000 to 2017. The complete dataset for NEH consists of 32,655 ratings for 80 ski runs that were assessed on 429 days during the five winter seasons from 2013 to 2017. Hence, each of the ski runs included in our analysis is characterized by a multi-season time series of daily run list ratings.

Since large datasets with many attributes are challenging for traditional clustering techniques (Assent, 2012), we applied a
two-step approach that combines the strengths and efficiency of self-organizing maps (SOM, Kohonen, 2001), an unsupervised competitive neural network clustering algorithm, with the transparency of traditional hierarchical clustering (Vesanto and Alhoniemi, 2000; Gonçalves et al., 2008). This approach circumvents the challenge of the large dataset by first using SOM to produce an analysis dataset with substantially fewer items that represent meaningful averages and are less sensitive to random variations than the run list time series included in the original data. Hierarchical clustering is subsequently applied to the
reduced dataset to derive the final groups of runs (Vesanto and Alhoniemi, 2000). While it would be possible to group the runs entirely with SOM, the dendrogram of hierarchical clustering allows a more transparent evaluation of the clustering solution. Vesanto and Alhoniemi (2000) showed that for large datasets this two-level clustering approach performs well compared with direct clustering.

SOM (Kohonen, 1982, 2001) is a machine learning algorithm that is particularly adept at pattern recognition and clustering in
large complex datasets (Kohonen, 2013). The method performs a nonlinear projection from the high-dimensional input data space to a smaller number of neural network nodes on a two-dimensional grid while preserving the topological relationships of the input data. SOM has been widely used as an analytical and visualization tool in exploratory and statistical data analysis in science and industrial applications (e.g., Kaski et al., 1998; Oja et al., 2003; Gonçalves et al., 2008; Pöllä et al., 2009; Radić et al., 2015; Shandro and Haegeli, 2018).

The neural network of a SOM consists of an input layer of $x$ $p$-dimensional observations and an output layer of $k$ neural nodes, each of which is characterized with a $p$-dimensional weight vector $w$ representing an archetypal pattern in the input data. In our case, the input data consists of time series of daily run list ratings for each run and the weight vectors of the SOM nodes represent typical time series of how those runs were coded. Each SOM node has a position on a two-dimensional map and an initial weight vector $w$ based on a randomly selected object from the input data. Training the network is performed for a chosen
number of iterations where the entire input dataset is presented to the network repeatedly. For each input vector the node with the closest weight vector—known as the "best matching unit" (BMU)—is individually determined using a specified distance measure. The network learns (i.e., "self-organizes") by adapting the weight vectors of the BMU and the nodes within a predefined neighborhood of the BMU to the input vector. This updating step is described by $w(t+1) = w(t) + \Theta(t)\alpha(t)[v(t) - w(t)]$, where $t$ is the current iteration, $w$ is the weight vector, $v$ is the input vector, $\Theta$ is the neighborhood



function that considers distance from the best matching node, and α is an iteration-dependent learning function. An essential characteristic of the SOM is that this iterative process eventually stabilizes in such a way that nodes that are similar to one another are situated close together on the map, thus preserving the topology of the input data. After the training process, individual SOM nodes represent archetypal patterns found in the original data. In our case, the patterns are characteristic time

series of run list ratings for the runs included in each node. The amount of original information retained depends primarily on the size of the SOM (i.e., the number of nodes), with smaller sizes producing broader generalizations of the input datasets and larger sizes capturing increasingly fine details. Following the work of Liu et al. (2006), we selected a map size that optimizes the average distance between each input vector (quantization error), minimizes the percentage of input vectors for which the first best matching unit and second best matching unit are not neighboring nodes (topographical error) and the percentage of

empty nodes on the map. Interested readers are referred to Kohonen (2001) for a more in-depth explanation of the development and details of the SOM algorithm.

To derive the final groups of runs, we applied hierarchical clustering to the characteristic run list rating time series identified by the SOM. Hierarchical clustering groups similar objects into clusters where each cluster is distinct from every other cluster, and the objects within each cluster are most similar to each other (Hastie et al., 2009). The main output of hierarchical clustering

is a dendrogram, which shows the hierarchical relationship between the clusters graphically. We chose the final number of groups of runs based on an inspection of the clustering dendrogram, while balancing resolution and interpretability of the cluster solution. Finally, we arranged the identified groups into a hierarchy by ordering them according to the average percentage of days the runs were open within each group.

To ensure that we extract meaningful patterns from our dataset, we preprocessed our input data prior to the clustering analysis

using the following steps. First, we needed to make the run list ratings of the two operations consistent. While the guides at CMHGL open or close entire runs, NEH rates the individual ski lines on each run in their run list. To make the analysis comparable between the two operations, we converted the NEH ski line ratings into run-level ratings by considering a run open as soon as at least one of its ski lines was open. Second, we excluded ski runs that were closed during the entire study period (e.g., ski runs that were kept in the run list as a reminder for the guiding team that a ski run is permanently closed due to

wildlife concerns) since these runs would not contribute any meaningful information to our analysis. Third, we only included ski runs in our analysis that were at least occasionally used. Following the recommendations of our collaborating senior guides, we only included runs that were skied at least once a season at NEH, while we restricted our CMHGL dataset to runs that were skied at least once during the entire study period. Fourth, we restricted the dataset to ski runs that were included in the run list of all winters of the study period (2013 to 2017 at NEH; 2007 to 2017 at CMHGL) since the employed clustering algorithms

are sensitive to large amounts of missing data. The final dataset for the SOM analysis consisted of 25,311 daily run list ratings from 59 ski runs on 429 days for NEH and 286,008 daily run list ratings from 227 ski runs on 1,260 days for CMHGL.

Since SOM requires input data to either be numerical or binary (i.e., 0 or 1), we had to recode our categorical run list ratings before processing. Following the approach of dummy coding routinely used for categorical data in regression analysis, we converted our original time series with five run list codes into two simplified binary time series. The first binary time series





describes whether a ski run was open with 1 representing the original run list codes *"Open for guiding"* and *"Conditionally open for guiding"* (CMHGL only). The second binary time series describes whether a ski run was closed for avalanche hazard with 1 standing for *"Closed for guiding"*. This means that runs that were open for guiding were coded as 1-0 (first binary time series – second binary time series), runs that were closed for avalanche hazard were coded as 0-1, and runs that were closed

for other reasons (*"Not discussed"*, *"Closed for guiding for other reasons than avalanche hazard"* (NEH only) or days with missing data) were coded as 0-0. The two binary time series of each run were then combined to produce the input data for the SOM analysis that represents the originally categorical nature of the run list data in a binary format. At the end of the training process of the SOM, the initially binary input data is represented by the weight vectors of each nodes as typical time series on a continuous scale between 0 and 1 that allows for the subsequent clustering with an appropriate similarity measure.

We performed our analysis using the R statistical software (R Core Team, 2017) and the Kohonen package (Wehrens and Buydens, 2007). We used a training length of 200 iterations, the Tanimoto similarity measure for binary data, a hexagonal topology, a circular neighborhood function and a decreasing learning rate from 0.05 to 0.01. For the subsequent hierarchical clustering we used Ward's minimum variance method appropriate for numerical data.

## 2.3    Characterization of the identified run groups

To understand the nature of the emerging groups of ski runs, we had a senior lead guide in each operation complete a detailed terrain characterization survey for all the runs included in our study. The collaborating guides had 20 and 34 years of guiding experience in mechanized skiing and guided at their operation for 5 years as the operations manager and 17 years as a lead guide respectively. The objective was to collect information on key characteristics that affect guiding teams to either open or close ski runs. While existing terrain studies have primarily focused on hazard information, we aimed for a more

comprehensive assessment that includes information on *Access*, *Type of Terrain*, *Skiing Experience*, *Operational Role*, *Hazard Potential*, and *Guide-ability* (see Table S1 for details on each run attribute and levels included). Each of these themes was assessed with a series of questions that asked about the presence or absence of specific features (e.g., "What type(s) of skiing terrain does this run include?"), included ordinal assessments of the magnitude or severity of features (e.g., "What is the steepness of the most serious slopes on this run?") and qualitative evaluations of the overall perception of the nature of the

terrain (e.g., "In terms of hazards, what is your sense of the overall friendliness of the terrain of this run?"). The last type of question aimed to capture the overall feel for the terrain that experienced guides develop based on their overall knowledge and experience with a ski run. We deliberately chose to mainly focus on guides' subjective assessment of the terrain instead of the more objective terrain parameters typically included in avalanche terrain studies (e.g., incline, aspect) since this information is much richer, and it is ultimately guides' subjective perception that drives their terrain choices.

The different themes aim to capture various aspects of operational decision-making. An important operational factor in helicopter-skiing is the ease of access of landings and pickups. *Access* captures the general accessibility with respect to required flying conditions as well as particular characteristics of the pickup location(s) such as overhead hazards which might limit accessibility of the ski run. *Type of Terrain* describes important terrain features and aims to capture the overall character of the





terrain of a ski run. Examples of the descriptors used for characterizing the type of terrain include glaciated alpine terrain, open slopes at tree line, open canopy snow forest (where the crowns of individual mature trees do not overlap), or large avalanche paths from above. Mechanized skiing operators aim to provide guests with an excellent skiing product and each ski run in their tenure offers certain operational benefits for achieving that. The theme *Skiing Experience* covers information on the overall

skiing experience and skiing difficulty level. *Operational Role* describes how a ski run is typically used in the ski program of the operation. While some ski runs can be used under almost all circumstances (i.e., "safe and accessible"), others are important jump runs that offer important connections among other ski runs and make daily circuits work. *Hazard Potential* aims to capture the relevant hazards of a ski run and was characterized in detail by individually assessing steepness, exposure, avalanche terrain hazards (e.g., avalanche overhead hazard to the ski line(s) or unavoidable unsupported terrain shapes), and

other hazards (e.g., crevasse or tree well hazard). For ski runs that were moderately steep or steeper, exposure was assessed by specifying the size of potential avalanche slopes (e.g., large avalanche slope(s) producing size 3.0 or larger). In addition, the overall friendliness of the terrain was assessed on a five-point Likert scale ranging from very friendly to very unfriendly. *Guide-ability* of a ski run describes how challenging it is to guide a group of guests safely through the terrain of that ski run (e.g., the terrain naturally leads guests to the right line or it requires detailed instructions and a close eye on the guest). This

aspect of a run was assessed using a four-point Likert scale including very easy, easy, difficult and very difficult.

The comprehensive run characterizations were summarized to describe the nature of the identified groups of runs. Specifically, we compared attribute frequencies of each group with overall attribute frequency among all ski runs of each operation. Because some groups of runs only contained relatively small numbers of runs, we focused on a more qualitative description of the nature of the groups instead of performing any statistical tests to compare groups.

**2.4    Seasonal variability in run groups**

To examine how the specific nature of individual winters might affect the grouping of ski runs, we applied hierarchical clustering for a second time. This time, we focused on individual seasons and clustered the representative time series of the previously identified groups to find groups of ski runs with similar run list rating patterns within that season and combined them into single groups. We chose the number of seasonal clusters based on an inspection of the clustering dendrogram using

Ward's minimum variance method.

**3    Results**

**3.1    Operational terrain classes at NEH**

**3.1.1    Run groups and overall terrain hierarchy**

For NEH, our analysis identified six groups of ski runs that exhibited distinct patterns in their run list ratings over the entire

period 2013 to 2017 (Figure 1a). After training several SOMs with varying number of nodes, we selected a robust SOM





solution with 6x3 nodes that optimizes the quantization and topographical errors. Based on the visualization of the node dissimilarities in the clustering dendrogram we finally chose a solution with six group of ski runs.

Figure 1a shows the NEH time series of run list ratings of consecutive winters (December 1 to March 31) grouped into the six identified groups. The time series strips of each group consist of colour-coded rows representing the run list ratings of the individual runs included in that group. Taller strips therefore represent groups with larger numbers of runs. Days when ski runs were open are shown in green, days when they were closed due to avalanche hazard are shown in red, and days when they were not discussed or closed due to non-avalanche hazard related reasons are shown in black. Days with no run list data at all (e.g., prior to operating season, days when operation was shut down due to inclement weather conditions) are shown in grey. A visual inspection of Figure 1a confirms the grouping of the runs as one can see considerable consistency in the run list rating patterns within groups. At the same time, we also find individual days when certain ski runs were coded differently than the rest of their group.

The groups of ski runs are arranged hierarchically according to the average percentage of days the runs in the group were open for skiing with guests over the five seasons. The group of runs shown at the very top was open for skiing with guests the most often with an average of 97% of the days during the study period (seasonal values ranging between 94% and >99%, Table S2). They were closed due to avalanche hazard on only 1% of the days and either not discussed or closed due to other reasons than avalanche hazard on 2% of the days. In contrast, the lowest group in the terrain hierarchy includes fourteen ski runs that were, on average, only open on 29% of the days during the study period (seasonal values ranging between 18% and 35%, Table S2). These runs were closed due to avalanche hazard on 61% of the days of a season and either closed due to other reasons than avalanche hazards or not discussed at all on 10% of the days.

### 3.1.2 Run group characterization

Based on the run characterization provided by our experienced guide contact, the skiing terrain of NEH generally offers a variety of skiing at all three elevation bands (Table 1). The majority of the 59 ski runs include non-glaciated alpine terrain and many comprise of open slopes at tree line or glades. However, the terrain at NEH also includes ski runs that go through open canopy snow forests below tree line. A fifth of all the ski runs include large avalanches paths formed from above. The majority of the ski runs were characterized as gentle or moderately steep. While sustained steep ski runs with exposure to large avalanches slopes capable of producing Size 3.0 avalanches exist, approximately half of the ski runs included in our study do not involve exposure to avalanches slopes.

Group 1, which consists of eight ski runs that are most frequently open, is characterized by mostly gentle terrain with ski lines that have none or only limited exposure to avalanche slopes (Table 1). Much of the ski terrain consists of open slopes at tree line or open canopy snow forest below tree line as well a few non-glaciated and glaciated alpine runs. The ski runs of this group provide easy skiing and generally a good skiing experience. Overall, the majority of the ski runs were characterized as safe and accessible under most conditions and many were identified as high efficiency production runs. At the same time, one



of the ski runs included in this group was flagged as only rarely being used because it provides a poor skiing experience for guests.

Group 2 is made up of nine gentle ski runs with no exposure to avalanche slopes on the ski lines. Another main feature of this group is that their terrain mainly consists of open slopes or glades at tree line. These runs are almost always accessible. While they provide easy skiing, the overall skiing experience was characterized as fair.

Group 3 consists of two runs and only includes ski runs that are always accessible and provide fair and good skiing through snow forest, glades and a large avalanche path formed from above. One ski run is moderately steep with short steep pitches and the ski line is exposed to multiple smaller avalanche slopes, while the other ski run is gentle with no exposure to avalanche slopes. Skiing is moderately challenging or challenging and guide-ability was characterized as difficult on one run and easy on the other.

While most of the ski runs of the first three groups are below or around tree line, the next three groups predominantly consist of alpine terrain. Group 4 consists of thirteen ski runs. The main characteristic of this group its gentle, non-glaciated or glaciated alpine terrain or its open slopes at tree line where most ski lines do not cross any avalanche slopes. These friendly or very friendly ski runs are often accessible and provide generally good skiing experience with easy or moderately challenging skiing. Some of the ski runs in this group can be exposed to overhead avalanche hazards during regular avalanches cycles (i.e., avalanche cycles producing avalanches up to Size 3.0).

All the thirteen ski runs of Group 5 are located in the alpine, many also include skiing on glaciers or through open slopes at tree line. Most of the ski runs are moderately steep or steeper and include travelling through smaller or large avalanche slopes. Almost half of the ski lines can be directly affected by overhead hazard during regular avalanches cycles, which makes this group exhibit the highest prevalence of that particular hazard. While the majority of the runs included in this group can be accessed by helicopter under most conditions, many pickup locations are threatened by overhead avalanche hazard during large avalanche cycles (producing avalanches of Size 3.5 or larger) and some of the pickups are even threatened during regular avalanche cycles. Many of the pickups are also exposed to the persistent presence of triggers for overhead hazards (e.g., ice fall or cornices). While skiing on these runs was mainly characterized as moderately challenging, they offer very good or even "life-changing" skiing experiences for guests. This group of runs is critical for the operation as many of the runs are high-efficiency production runs, and numerous runs are used as a destination in a daily skiing program or are perceived as providing a skiing experience that defines the operation.

Group 6 mainly includes moderately challenging or challenging alpine ski runs that are rarely skied but can play an important operation role under special circumstances and runs that are only considered under "bomb-proof" conditions. Most of these fourteen ski runs have moderately steep or steeper slopes that can produce avalanches of Size 3.0 or bigger. Many pickups locations are regularly exposed to overhead avalanche hazard. However, ski runs in this group provide good or very good skiing experiences for guests.



Figure 1: Identified terrain hierarchy with groups of similarly managed ski runs at NEH with (a) typical time series of run list ratings for the winter seasons 2013 to 2017 and (b) inter-seasonal variation within the terrain hierarchy. Open ski runs are shown in green and ski runs closed due to avalanche hazard are shown in red. Ski runs that were not discussed or closed to other reasons than avalanche hazard are shown in black. Days with no data are shown in grey.


**Table 1: Characteristics of the identified terrain groups at NEH and CMHGL (percentages that are greater than the basic distribution across all groups at an operation are highlighted with bold font and shaded in grey).**

| Attribute and levels | | | Group at NEH | | | | | | | Group at CMHGL | | | | | | |
|---|---|---|---|---|---|---|---|---|---|---|---|---|---|---|---|---|
| | | All | 1 | 2 | 3 | 4 | 5 | 6 | All | 1 | 2 | 3 | 4 | 5 | 6 | 7 |
| Number of ski runs | | 59 | 8 | 9 | 2 | 13 | 13 | 14 | 227 | 44 | 38 | 48 | 12 | 31 | 21 | 33 |
| **Access** | | | | | | | | | | | | | | | | |
| Required flying conditions | | | | | | | | | | | | | | | | |
| *Run is almost always accessible* | | 28 | 38 | 56 | 100 | 17 | 15 | 14 | 28 | 64 | 58 | 23 | 8 | 3 | - | - |
| *Run is often accessible* | | 60 | 63 | 44 | - | 67 | 85 | 50 | 22 | 20 | 26 | 35 | 25 | 19 | 10 | 9 |
| *Conditions must line up* | | 10 | - | - | - | 8 | - | 36 | 31 | 14 | 16 | 38 | 42 | 48 | 57 | 27 |
| *Conditions must be perfect* | | 2 | - | - | - | 8 | - | - | 19 | 2 | - | 4 | 25 | 29 | 33 | 64 |
| Particular pickup features | | | | | | | | | | | | | | | | |
| *Overhead hazard, regular avalanche cycles* | | 9 | - | - | - | - | 15 | 21 | 20 | 7 | - | 21 | 50 | 16 | 43 | 39 |
| *Overhead hazard, large avalanche cycles only* | | 47 | 13 | 22 | 50 | 42 | 69 | 64 | 53 | 25 | 63 | 54 | 42 | 77 | 52 | 61 |
| *Common trigger for overhead hazard* | | 10 | - | - | - | - | 38 | 7 | - | - | - | - | - | - | - | - |
| **Type of terrain**[a] | | | | | | | | | | | | | | | | |
| *Extreme alpine faces* | | 2 | - | - | - | - | 8 | - | 2 | - | - | - | - | - | - | 12 |
| *Glaciated alpine* | | 26 | 25 | - | - | 50 | 38 | 14 | 11 | 2 | 3 | 2 | 8 | 19 | 19 | 33 |
| *Non-glaciated alpine* | | 66 | 38 | - | - | 75 | 100 | 93 | 7 | 7 | 3 | 8 | 17 | 6 | 5 | 12 |
| *Open slopes at tree line* | | 41 | 50 | 67 | 50 | 50 | 46 | 7 | 68 | 39 | 39 | 65 | 67 | 97 | 100 | 97 |
| *Glades* | | 38 | 25 | 89 | 100 | 33 | 31 | 14 | 17 | 7 | 50 | 17 | 8 | 10 | 14 | 3 |
| *Open canopy snow forest* | | 17 | 38 | 44 | 100 | - | - | 7 | 29 | 66 | 58 | 21 | - | 3 | 10 | 3 |
| *Dense forest* | | 2 | - | 11 | - | - | - | - | 1 | 7 | - | - | - | - | - | - |
| *Cut blocks* | | 3 | 13 | 11 | - | - | - | - | 3 | 9 | 5 | - | - | - | 5 | - |
| *Large avalanche path formed from above* | | 21 | - | 11 | 50 | 8 | 38 | 29 | 56 | 9 | 32 | 69 | 67 | 81 | 81 | 82 |
| *Planar slopes* | | 9 | - | 11 | - | - | 15 | 14 | 3 | - | - | - | - | 10 | - | 12 |
| **Skiing experience** | | | | | | | | | | | | | | | | |
| Skiing difficulty | | | | | | | | | | | | | | | | |
| *Easy* | | 33 | 63 | 56 | - | 58 | 15 | - | 16 | 34 | 13 | 23 | 25 | 6 | - | 3 |
| *Moderate* | | 50 | 38 | 33 | 50 | 42 | 69 | 57 | 71 | 66 | 76 | 58 | 67 | 87 | 95 | 61 |
| *Challenging* | | 17 | - | 11 | 50 | - | 15 | 43 | 13 | - | 11 | 19 | 8 | 6 | 5 | 36 |
| Overall guest experience | | | | | | | | | | | | | | | | |
| *Poor (Happy to move on)* | | 7 | 13 | 22 | - | - | - | 7 | 4 | 16 | - | 4 | - | - | - | - |
| *Fair (Not bad skiing)* | | 21 | 13 | 44 | 50 | 17 | 15 | 14 | 17 | 25 | 21 | 21 | 25 | 6 | 19 | 3 |
| *Good (A good product)* | | 41 | 75 | 33 | 50 | 67 | 8 | 36 | 37 | 43 | 42 | 31 | 58 | 48 | 24 | 24 |
| *Very good (This is why guests come back for more)* | | 26 | - | - | - | 17 | 62 | 36 | 33 | 16 | 32 | 38 | 8 | 35 | 52 | 42 |
| *Exceptional (Life changing mountain experience)* | | 5 | - | - | - | - | 15 | 7 | 9 | - | 5 | 6 | 8 | 10 | 5 | 30 |
| **Operational role** | | | | | | | | | | | | | | | | |
| *Safe and accessible under almost all conditions* | | 41 | 88 | 78 | 100 | 58 | 8 | - | 6 | 30 | - | - | - | - | - | - |
| *Bread and butter (high efficiency production run)* | | 33 | 38 | 33 | 50 | 25 | 54 | 14 | 19 | 59 | 37 | 6 | - | - | - | - |
| *Key jump run (makes a circuit work)* | | 28 | 38 | 44 | 50 | 42 | 23 | - | 5 | 9 | 5 | 4 | - | 3 | 10 | - |
| *Regular lunch run* | | 9 | 25 | - | - | 8 | 15 | - | 4 | 14 | 3 | 2 | - | - | - | - |
| *Time management run (e.g., used during fuel run of helicopter)* | | 5 | - | 22 | - | - | 8 | - | - | - | - | - | - | - | - | - |
| *Destination run (objective of a circuit)* | | 12 | - | - | - | 17 | 31 | 7 | 6 | - | - | 10 | 17 | - | 14 | 9 |
| *Signature run (defining the operation)* | | 7 | - | - | - | - | 23 | 7 | 1 | 2 | 3 | - | - | - | - | - |
| *Open season run (only considered under bomb-proof conditions)* | | 10 | - | - | - | - | 8 | 36 | 2 | - | - | - | 3 | - | - | 9 |
| *Rarely skied (but important under special circumstances)* | | 24 | 13 | - | - | 17 | 23 | 57 | 2 | - | - | - | - | - | - | 12 |
| *Not preferred run (considered when lacking reasonable skiing)* | | 10 | - | 22 | - | 8 | - | 21 | 4 | 14 | - | 2 | - | - | 5 | - |
| **Hazard potential** | | | | | | | | | | | | | | | | |
| Steepness | | | | | | | | | | | | | | | | |
| *Gentle* | | 47 | 75 | 100 | 50 | 75 | 15 | - | 2 | 11 | - | - | - | - | - | - |
| *Moderate* | | 28 | 25 | - | - | 25 | 46 | 36 | 11 | 20 | 18 | 10 | 8 | 6 | 5 | - |
| *Moderate with steep pitches* | | 14 | - | - | 50 | - | 23 | 29 | 43 | 52 | 63 | 52 | 67 | 23 | 33 | 12 |
| *Sustained steep* | | 12 | - | - | - | - | 15 | 36 | 44 | 16 | 18 | 38 | 25 | 71 | 62 | 88 |





| Attribute and levels | Number of ski runs | Group at NEH | | | | | | | Group at CMHGL | | | | | | | |
|---|---|---|---|---|---|---|---|---|---|---|---|---|---|---|---|---|
| | | All | 1 | 2 | 3 | 4 | 5 | 6 | All | 1 | 2 | 3 | 4 | 5 | 6 | 7 |
| Number of ski runs | | 59 | 8 | 9 | 2 | 13 | 13 | 14 | 227 | 44 | 38 | 48 | 12 | 31 | 21 | 33 |
| **Hazard potential (continued)** | | | | | | | | | | | | | | | | |
| Exposure to avalanche slopes on the ski line(s) | | | | | | | | | | | | | | | | |
| *None* | | 47 | **75** | **100** | **50** | **75** | 15 | - | 1 | **5** | - | - | - | - | - | - |
| *Single small slopes, can produce Size ≤ 2.5 avalanches* | | 12 | **13** | - | - | 17 | 23 | 7 | 5 | **7** | 5 | 8 | 17 | - | 5 | - |
| *Multiple small slopes, can produce Size ≤ 2.5 avalanches* | | 19 | 13 | - | 50 | - | 31 | 36 | 44 | **82** | **79** | 44 | 50 | 13 | 10 | - |
| *Large slope(s), can produce Size ≥ 3.0 avalanches* | | 22 | - | - | - | 8 | 31 | 57 | 50 | 7 | 16 | 48 | 33 | **87** | **86** | **100** |
| Avalanche terrain hazards[b] | | | | | | | | | | | | | | | | |
| *Overhead hazard, regular avalanche cycles* | | 16 | - | - | - | 17 | 46 | 7 | 24 | 9 | 5 | 29 | 33 | 29 | 48 | 36 |
| *Overhead hazard, large avalanche cycles only* | | 10 | 13 | 22 | - | 25 | - | - | 22 | 14 | 21 | 15 | 33 | 19 | 38 | 30 |
| *Common trigger for overhead hazard* | | 3 | - | - | - | 8 | 8 | - | 1 | - | 3 | - | - | - | 5 | 3 |
| *Unavoidable unsupported terrain shapes* | | 7 | - | - | - | 17 | 8 | 7 | 2 | - | - | 2 | - | 3 | - | 9 |
| *High consequence terrain* | | 3 | - | - | - | - | 8 | 7 | 2 | - | - | 2 | - | 3 | - | 9 |
| Other hazards[b] | | | | | | | | | | | | | | | | |
| *Crevasse hazard, isolated* | | 9 | - | - | - | 8 | 23 | 7 | 4 | - | 3 | 2 | 8 | 3 | 5 | 9 |
| *Crevasse hazard, widespread and/or unavoidable* | | 2 | - | - | - | - | - | 7 | 2 | - | - | 2 | - | - | - | 12 |
| *Cornices directly affecting the ski line(s)* | | 12 | - | - | - | - | 31 | 21 | 5 | - | - | 4 | - | 16 | 14 | 6 |
| *Tree well hazard* | | 9 | - | 22 | 50 | 8 | - | 7 | 4 | 16 | - | 2 | - | - | - | - |
| *Open creeks, vent holes etc.* | | 3 | 13 | - | - | - | 8 | - | 3 | - | 3 | 6 | - | 3 | - | 6 |
| Overall friendliness | | | | | | | | | | | | | | | | |
| *Very friendly* | | 34 | **88** | **44** | - | **58** | 15 | - | 6 | **25** | 3 | 2 | - | - | - | - |
| *Friendly* | | 19 | 13 | 11 | 50 | 42 | 8 | 14 | 21 | **61** | **32** | 13 | 8 | 3 | 5 | - |
| *Neutral* | | 26 | - | 44 | 50 | - | 46 | 29 | 19 | 9 | 34 | 19 | 58 | 23 | - | 6 |
| *Unfriendly* | | 16 | - | - | - | - | 23 | 43 | 43 | 5 | 32 | 65 | 33 | 55 | 81 | 45 |
| *Very unfriendly* | | 5 | - | - | - | - | 8 | 14 | 11 | - | - | 2 | - | 19 | 10 | 48 |
| **Guide-ability** | | | | | | | | | | | | | | | | |
| *Very easy* | | 39 | **50** | 11 | - | **50** | 38 | **50** | 5 | **7** | 3 | 4 | - | 13 | 5 | - |
| *Easy* | | 37 | **50** | 33 | 50 | 42 | 46 | 21 | 42 | 32 | 39 | 33 | 50 | 45 | 52 | 61 |
| *Difficult* | | 22 | - | 56 | 50 | 8 | 15 | 29 | 52 | 59 | 58 | 63 | 50 | 35 | 43 | 39 |
| *Very difficult* | | - | - | - | - | - | - | - | 1 | **2** | - | - | - | 6 | - | - |

[a] Only the ten most prominent types of terrain in both operations are shown.

[b] Only the five most prominent avalanche terrain hazards resp. other hazards in both operations are shown.

### 3.1.3 Inter-seasonal variations

The seasonal clustering of the long-term terrain groups discussed above revealed that for some of the seasons groups of runs adjacent to each other on the terrain hierarchy could be combined as they are coded very similarly during a season (Figure 1b).

5 While the identified long-term terrain hierarchy consists of six groups, the number of seasonal groups ranges from four to six with an average of five groups per season. This additional seasonal grouping was only observed among the first three groups where most ski runs are at tree line or below. Groups 1 and 2 were combined for three times (2013, 2016 and 2017) out of the five seasons. Similarly, Groups 2 and 3 were coded very similarly during the seasons 2013, 2015 and 2016. On the other hand, Groups 4, 5 and 6 had more distinct run list rating patterns during all five seasons. These three groups, which mainly consist

10 of alpine ski runs, were never clustered together.



## 3.2 Operational terrain classes at CMHGL

### 3.2.1 Run groups and overall terrain hierarchy

For CMHGL, our analysis identified seven groups of ski runs that were coded similarly over the entire study period from 2007 to 2017 (Figure 2a). In this case, a SOM solution with 6x5 nodes optimized the quantization and topographical errors and the resulting 30 archetype patterns were subsequently used as input for the hierarchical clustering. Based on the visualization of the node dissimilarities in the clustering dendrogram we chose a final solution with seven clusters.

At the top of CMHGL's terrain hierarchy is Group 1, which includes 44 ski runs that were almost always open. Over the entire study period, these ski runs were open for skiing with guests on 93% of the days (seasonal values ranging between 86% and 98%, Table S3). They were closed due to avalanche hazard on only 3% of the days and either not discussed or closed due to other reasons than avalanche hazard on 4% of the days. At the other end of spectrum, the lowest group in the identified terrain hierarchy consists of 33 ski runs that were only open on 16% of the days (seasonal values ranging between 5% and 32%). These runs were closed due to avalanche hazard on 67% of the days and not discussed at all on 17% of the days.

### 3.2.2 Group characterization

The overall character of the ski terrain at CMHGL is dominated by steep tree skiing. While some runs start in the alpine, the vast majority of the 227 ski runs involves skiing through open slopes at tree line or open canopy snow forest below tree line. More than half of all the ski runs involve skiing through large avalanche paths formed from above. Most of the ski runs were characterized as either moderately steep but with steep pitches or as sustained steep. Many runs involve skiing with exposure to multiples small slopes capable of producing up to Size 2.5 avalanches or even to large slopes that can produce avalanches of Size 3.0 or greater.

The ski runs in the first three groups at CMHGL are predominantly located at tree line or below. The ski terrain of the 44 ski runs in Group 1 is characterized mainly as snow forest with open canopy, dense forest or cut blocks. However, a few runs contain open slopes at tree line and both non-glaciated or glaciated sections in the alpine. Most of the ski runs are moderately steep, but half of them include steep pitches. Most of these ski runs involve exposure to multiple small avalanche slopes that can produce avalanches up to Size 2.5. Many ski runs in Group 1 provide good skiing experience and most them are almost always accessible. Overall, the terrain in this group is predominantly characterized as friendly and the ski runs are either high-efficiency production runs or runs that are safe and accessible under most conditions.

Group 2 includes 38 almost always accessible ski runs where the terrain is similar to the runs included in Group 1—open canopy snow forests and cut blocks at and below tree line—but also features more glades and more large avalanche paths formed from above. Most of the ski runs are moderately steep but include steep pitches with exposure to multiple small avalanche slopes that can produce avalanches up to Size 2.5. The friendliness of the ski runs in this group ranges from friendly to unfriendly, but most of them are perceived in the middle as neither friendly or unfriendly. The ski runs in Group 2 mainly provide good skiing experience and their operational roles are mainly high-efficiency production runs.



Group 3, the biggest group in the CMHGL terrain hierarchy, consists of 48 ski runs that mainly have steep pitches or are sustained steep on open slopes at tree line. Skiing involves exposure either to multiple small or even large avalanche slopes on the ski lines and a third of the ski runs includes exposure to overhead hazard during regular avalanche cycles. Moreover, Group 3 is the first group with a substantial proportion of runs that require skiing through avalanche paths formed from above.

While the runs included in this group cover the full range of perceived friendliness, most of them are perceived as being unfriendly. The ski runs of this group are considerably less accessible than the runs of the previous groups and approximately one fifth of the pickup locations can be exposed to overhead hazard during regular avalanche cycles. However, many of these ski runs provide very good skiing experiences.

Group 4 consists of twelve ski runs that offer similar terrain as Group 3. However, these ski runs are even less accessible than

the runs of Group 3, and half of the pickup locations can be exposed to overhead hazard during regular avalanche cycles. The ski runs are predominantly moderately steep but include steep pitches and multiple smaller avalanche slopes. In addition to open slopes at tree line and many large avalanche paths, some of these ski runs include non-glaciated or glaciated alpine terrain with isolated crevasse hazard. Overall, the friendliness of these ski runs is predominantly perceived as neutral. Most of these ski runs provide a good skiing experience and are mainly used as a destination of a daily skiing circuit.

The three groups at the bottom of CMHGL's terrain hierarchy all consist of ski runs at tree line or above that also contain substantial glaciated sections. The ski runs of these three groups are predominantly sustained steep and skiers are mainly exposed to large slopes capable of producing avalanche of Size 3.0 or bigger. In Group 5, the vast majority of the 31 ski runs are sustained steep and include large avalanche slopes. Almost all these ski runs include open slopes at tree line, large avalanche paths and involve some glaciated alpine terrain. Many of the ski lines on these runs are exposed to overhead avalanche hazard

during regular avalanches cycles and some have the potential of being hit by cornices from above. Most of these ski runs are perceived as unfriendly, but they provide good skiing. Generally, accessing these ski runs required flight conditions to line up or even be perfect. However, only some pickup locations are exposed to overhead hazard during regular avalanche cycles.

Group 6 includes 21 ski runs that are mainly sustained steep with exposure to large avalanche slopes on the ski lines. Their terrain consists of open slopes at tree line, many large avalanche paths and some glaciated alpine terrain. Most prominently,

overhead hazard during regular avalanches cycles is a concern for almost half of the ski runs in this group. In addition, some of the ski runs have overhead cornices directly affecting the ski lines. This group of ski runs is perceived as unfriendly, but it provides very good skiing. Just like in Group 5, flight conditions need to line up or even be perfect for accessing these runs, but many of the pickup locations in Group 6 are also exposed to overhead hazard.

Group 7 offers the most severe, least accessible but also some of the best skiing terrain within the tenure of CMHGL. The 33

ski runs in this group are predominantly sustained steep and all of them involve skiing on slopes that can produce large avalanches of Size 3.0 or larger. Flying conditions must be perfect to consider the runs of this group and many of the pickup locations are threatened by avalanches during regular avalanche cycles. Besides skiing on open slopes at tree line and through large avalanche paths, both non-glaciated and glaciated alpine terrain, this is the only group of runs which includes extreme alpine faces. Most frequently mentioned hazards in this group are unavoidable and unsupported terrain shapes, high





consequence terrain when caught in an avalanche, and crevasse hazard (especially widespread and/or unavoidable). Overall, the ski runs in this group are characterized as very unfriendly. From an operational perspective, these ski runs represent destinations of a daily skiing program or are only considered when conditions bomb-proof. Even though many of these ski runs provide very good or even exceptional skiing, these runs are only rarely skied.

### 3.2.3 Inter-seasonal variations

The cluster analysis based on the typical seasonal time series shows that in most seasons several groups of runs were coded similarly (Figure 2b, seasonal groups indicated with black boxes). On average, the seasonal terrain hierarchy consists of five groups but ranges from only four to all seven groups that were identified over the entire period. While the seasonal clustering at NEH only revealed seasonal groupings at the top of the terrain hierarchy, the analysis at CMHGL showed seasonal groupings

at all levels. Groups 1 and 2 were grouped together three times (2009, 2016 and 2017) out of eleven seasons. Groups 2 and 3 had very similar seasonal run list coding patterns only in 2007 and 2012. On the other hand, Groups 3 and 4 showed strong similarities in how they are coded and were grouped together in five seasons (2008, 2010, 2015, 2016, 2017). These two groups of ski runs have similar characteristics in terms of skiing terrain and hazard potential on the ski run, but they differ in accessibility as the pickup locations in Group 4 are characterized as being more exposed to overhead avalanche hazards. The

step from Groups 4 to 5 emerges as a strong transition in the terrain hierarchy at CMHGL as these two groups were only combined once (2007). Nearly all the ski runs in Group 5 consist of sustained steep ski runs at tree line or in glaciated alpine with exposure to large avalanche slopes that can produce size 3.0 avalanches or bigger. Groups 5 and 6 have very similar run list coding patterns and were grouped together in six of the eleven seasons (2008, 2009, 2010, 2015, 2016, 2017). They offer very similar type of skiing terrain, but the pickup locations of Group 6 are characterized as being more exposed to overhead

avalanche hazard. The step between the two lowest groups in the CMHGL terrain hierarchy marks a second significant transition as they were consistently coded differently and only grouped together once (2015). Group 7 is the only group that contains ski runs that were either characterized as extreme alpine faces or have widespread/or unavoidable crevasses.



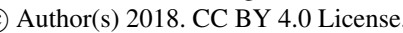


**Figure 2: Identified terrain hierarchy with groups of similarly managed ski runs at CMHGL with (a) typical time series of run list ratings for the winter seasons 2007 to 2017 and (b) inter-seasonal variation within the terrain hierarchy. Open ski runs are shown in green and ski runs closed due to avalanche hazard are shown in red. Ski runs that were not discussed or closed to other reasons than avalanche hazard are shown in black. Days with no data are shown in grey.**



## 4    Discussion

### 4.1    Customized terrain classes and terrain hierarchy

We identified distinct groups of ski runs based on run list ratings (i.e., revealed terrain preferences) that represent the avalanche risk management expertise of the local guiding teams. In comparison to existing terrain classification systems (e.g., ATES)
that divide terrain into a small number of universal classes, our analysis of run list ratings identifies high-resolution terrain hierarchies that offer a more detailed terrain description and reflect the variety and relative characteristics of available local terrain in a more meaningful way. The local nature of the terrain hierarchy is illustrated by the fact that the characteristics of the most frequently open groups of runs differ greatly between the two operations included in this study. At NEH, this group is predominantly characterized by gentle terrain with no exposure to avalanche slopes and includes ski runs in all elevation
bands. At CMHGL, the most frequently open group mainly consists of ski runs below tree line that include steep pitches and exposure to multiple small slopes capable of producing avalanches up to Size 2.5. We interpret this difference to reflect variations in the available terrain and operational practices at the two participating operations.

The terrain characteristics associated with the emerging terrain hierarchies generally agree with our existing understanding of what determines the severity of avalanche terrain (see, e.g., McClung and Schaerer, 2006; Statham et al., 2006). Both steepness
and size of the avalanche slopes skied emerged as strong drivers behind the observed terrain groups at both operations. The identified terrain hierarchies are also generally consistent with the nature of the terrain classes described in the ATES system (Statham et al., 2006). The ski runs that were less frequently open were generally characterized as having more unavoidable unsupported terrain shapes. Similarly, runs that were classified as more severe generally included more convoluted terrain and had more open planar slopes capable of producing large avalanches. The ski runs that were less frequently open were also
characterized more frequently as having high consequence terrain. Ski runs with large avalanche paths formed from above or overhead hazard during regular avalanche cycles were generally associated with groups that are less frequently open.

However, our analysis also revealed some notable differences that, at first glance, seem inconsistent with the established understanding of avalanche terrain severity. At NEH, the most obvious example is that the group of most frequently open ski runs contains runs that include glacier travel. In the ATES system, the presence of glaciated terrain automatically puts ski runs
into the most severe terrain class (Statham et al., 2006). Another example at NEH is Group 5, which includes a few runs without any avalanche related hazards on the ski line itself. However, these runs are often closed because the pickup locations can be affected by overhead avalanche hazard during regular avalanches cycles. At CMHGL, a noteworthy exception is Group 1, which contains seven ski runs below tree line that are sustained steep and have multiple slopes that can produce avalanches up to size 2.5. While the physical terrain characteristics of these runs would not necessarily suggest that they belong
into the group of runs that are open most often, the reason for their classification is the fact that they are actively maintained by the guiding team. Guides intentionally choose to ski these runs on a regular basis to destroy any potential weak layers before they are buried and become a risk management problem (R. Atkins, personal communication). This risk management practice





allows CMHGL to have these runs open more often than their physical terrain characteristics would suggest and ski steeper terrain than on unmanaged ski runs under similar hazard conditions.

These observations clearly demonstrate the ability of our approach to capture the nuanced terrain selection and risk management expertise of guides and turn them into insightful terrain hierarchies within local contexts. The groups of similarly types of ski runs reflect terrain severities at individual mechanized skiing operations in relation to the available terrain, local snow and avalanche climate and operational practices. Characterizing the identified groups with hazard considerations beyond the ones that just affect the ski lines (e.g., exposure of the pickup locations to overhead avalanche hazard) offers a more comprehensive description of their severity. This makes the derived terrain hierarchy more meaningful for operational use and the development of useful decision aids.

## 4.2 Seasonal variations in long-term operational terrain hierarchies

Our analysis of seasonal variation in terrain hierarchies highlights the necessity of long-term records for studying patterns in avalanche terrain selection in a meaningful way. While the overall structure of the terrain hierarchies was consistent throughout the entire study period, our within-season ski run group clustering revealed considerable season to season variabilities due to the specific meteorological character of a winter or particular sequences of weather events. At NEH, the observed seasonal variations illustrate the potential influence of the peculiar seasonal weather on avalanche conditions. While the first three groups of the terrain hierarchy at NEH are usually coded similarly, the ski runs in Group 2 were open on fewer days than average during the 2014 and 2015 winters (79% resp. 61% compared to 86%). Many regions in western Canada reported record low snowpack heights for the 2014 winter, and the warmer-than-usual 2015 winter was characterized by below average snowfall and well above average rainfall (SFU Avalanche Research Program, 2018). As a result, the lower elevation ski runs of Group 2 were not discussed or closed for other reasons than avalanche hazards (e.g., marginal snowpack, increased skiing hazards for the guests) more than a third of the days during the 2015 season. At the same time, the alpine ski runs of Groups 5 and 6 were open more than usual due the longer than usual fair-weather periods during that season and favorable avalanche conditions in the alpine.

At CMHGL, Groups 1 and 2 are usually coded differently, but they were managed more similarly during the winter seasons of 2009, 2016 and 2017. In 2009, the similarity is due to a major avalanche cycle that occurred in early January when most of the ski runs in both groups were closed for a few days. This cycle was due to the combination of a persistent weak layer buried early in December and one of the season's largest snowfalls. Many avalanches during this cycle ran to valley bottoms and, in some cases, beyond historical runout zones (SFU Avalanche Research Program, 2018). In 2016 and 2017, the similarity between the two groups was due to Group 2 ski runs being open considerably more often than normal because the forested and gladed terrain of Groups 1 and 2 ski runs was particular well suited for the conditions of these two seasons. The 2016 season started unseasonably warm with freezing levels reaching up to 2,300 m in December. The subsequent clear and stable conditions in early January produced a persistent weak interface in the snowpack that dominated the nature of avalanche hazard during that winter. The 2017 winter started with some of the season's coldest temperatures, unsettled conditions and continued





snowfall forming a mid-December interface that would remain a major feature of the snowpack for the rest of the season. The conditions during these two winters clearly favoured the use of the Groups 1 and 2 ski runs, which were consistently open throughout the season, while the runs of other groups were closed as soon as the early season interfaces were buried.

## 4.3 Additional factors affecting terrain hierarchies

In addition to offering insight on how avalanche hazard characteristics affect run list ratings, our analysis also highlights how non-avalanche hazard related factors affect the process and hence the revealed patterns. At NEH, for example, the ski run "Evil Twin Sister" was assigned to Group 5, which is open only about half of the time. While most ski runs in this group involve skiing through substantially severe avalanche terrain that is also exposed to overhead hazard, the "Evil Twin Sister" is a gentle ski run with no exposure to avalanche hazard. The reason for this discrepancy is likely the fact that the "Evil Twin Sister" only

provides a fair skiing experience and might therefore be discussed less frequently than other ski runs of similar terrain severity that offer better skiing experiences. In general, however, the quality of the skiing experience tends to correlate well with the terrain hierarchies that emerged at both participating operations. While the more severe ski runs at each operation are only rarely open, they are often described as offering exceptional skiing experience for guests. Our results at CMHGL show that the flying conditions required for accessing a run is also an important consideration during the run list rating process. Overall,

accessibility strongly decreases throughout the terrain hierarchy at CMHGL, and pickup locations that are threatened from above during regular avalanche cycles are a common concern in the run groups lower on the terrain hierarchy. Since our NEH analysis only included runs from their core operating area, this pattern did not emerge to a similar degree for NEH. However, it is typical that the runs located in drainages away from their core operating area are only discussed when the expected flying conditions allow guides to access these places in the first place (C. Israelson, personal communication). These examples

demonstrate that patterns in revealed terrain choices are the result of complex interactions between avalanche hazard factors and other operational considerations. While some of these patterns reflect natural collinearities (e.g., severity of avalanche terrain and ease of access), it is critical to consider non-avalanche related factors when interpreting patterns in revealed terrain choices and using the extracted knowledge for developing operational tools and decision aids.

## 4.4 Limitations

While our analysis offers valuable insight about the terrain hierarchy at the two participating operations, we acknowledge that our characterizations of the identified groups of ski runs were only based on the perspective of a single experienced guide. Since our characterizations not only included assessments of measurable physical characteristics, but also more intangible aspects and subjective assessments that integrate a wide variety of factors and personal experiences, it is reasonable to assume that these perspectives might vary among guides. However, the opening or closing of ski runs during the daily guides meeting

is a consensus-based group decision, and we believe that the opinions expressed by senior guides with extensive terrain experience under a wide variety of conditions likely carry more weight than the perspective of more junior guides. We therefore believe that the senior guides' assessments offer a valid characterization of the terrain that is sufficient for the present analysis.





## 5    Conclusions

We used multi-season datasets of daily run list ratings at two commercial backcountry skiing operations to identify groups of similarly treated ski runs and arrange them into operation-specific terrain hierarchies that reflect the local terrain expertise and avalanche risk management practices in the context of the available terrain and local snow and avalanche climate conditions.

To characterize the revealed terrain classes in detail, we had a senior lead guide at each operation describe the nature of each of the ski runs included in the study with respect to access, type of terrain, skiing experience, operational role, hazard potential, and guide-ability. While earlier studies exploring the terrain management expertise of mountain guides at the run scale were confined to hypothetical decision situations (Grimsdottir, 2004; Haegeli, 2010b), we present a flexible approach for identifying patterns in actual risk management decisions. To our knowledge, this is the first time that large operational backcountry skiing

datasets have been used to identify patterns in professional terrain selection and formally extract the operational avalanche risk management expertise at the run scale.

The results of our study offer numerous contributions for future research on avalanche risk management. A meaningful representation of terrain is critical for properly linking backcountry risk management decisions to avalanche hazard conditions. The terrain classes identified in our study therefore provide an exciting opportunity for exploring the link between run list

decisions and different avalanche hazard and weather situations. This type of analysis has the potential to offer useful insight into what type of terrain is acceptable under different avalanche hazard conditions and provide the foundation for developing evidence-based and condition-specific terrain guidance tools.

While revealed terrain preference data from GPS tracking units (e.g., Hendrikx et al., 2016; Thumlert and Haegeli, 2017) offer promising avenues for learning about professional avalanche risk management expertise at spatial scales below the run level,

it is important to remember that terrain decisions in mechanized skiing operations are made in stages (Israelson, 2013, 2015). Since small-scale terrain choices are only made within runs that were previously considered open for guiding, the patterns captured in the operation-specific terrain hierarchies presented in this study offer critical context for the meaningful analyses of GPS data. Furthermore, our study reiterates that it is difficult to relate terrain choices to physical terrain characteristics alone (Haegeli and Atkins, 2016). Examples of important other factors that emerged from our study include exposure of pickup

locations to overhead hazard, accessibility of ski runs, previous skiing on runs and the type and quality of the guest skiing experience. To identify insightful patterns and analytically isolate the effect of avalanche hazard, it is critical to examine revealed terrain preference data within the full array of influencing factors. Finally, the results of our study highlight that having long-term datasets is critical for identifying meaningful patterns in risk management practices as the particularities of individual winters can affect observed choices considerably.

More importantly, however, we believe that the terrain hierarchies produced by our analysis can overcome some of the challenges that have prevented the adoption of terrain classification systems in mechanized skiing operations in the past. While the categories of existing avalanche terrain classification system have been too broad and generic for providing meaningful insight, the larger number of terrain classes included in our terrain hierarchies and their operation specificity offer a much





more nuanced and applied perspective of the terrain. Since the patterns identified by our analysis reflect actual risk management practicies that have been used at participating operations for many years, the terrain hierarchies developed through our approach are much closer to the risk management decisions the classification aims to support than existing systems. Furthermore, the reflective nature of our approach and the fact that the emerging classification is grounded in past local risk

management decisions further increase guides' input into the system, which has the potential to increase acceptance and trust in the system. This means that the approach for the development of operation-specific, risk management focused terrain hierarchies presented in this study opens new opportunities for the development and application of meaningful risk management decision aids at mechanized skiing operations.

## 6    Acknowledgments

We would like to thank Canadian Mountain Holiday Galena and Northern Escape Heli Skiing for their willingness to participate in this study. Especially, we would like to thank Clair Israelson and Roger Atkins for their contribution to this work by taking the time and characterizing the ski runs in their tenures. The NSERC Industrial Research Chair in Avalanche Risk Management at Simon Fraser University is financially supported by Canadian Pacific Railway, HeliCat Canada, Mike Wiegele Helicopter Skiing, the Canadian Avalanche Association. The research program receives additional support from Avalanche

Canada and the Avalanche Canada Foundation. Reto Sterchi was also supported by SFU's Big Data Initiative KEY and a Mitacs Accelerate fellowship in partnership with HeliCat Canada.

## 7    Data availability

The operational data used in this study are confidential and cannot be made publicly available. Interested parties may wish to contact the corresponding author for further information.

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
