# Peer review of "A method of deriving operation-specific ski run classes for avalanche risk management decisions in mechanized skiing"

_Natural Hazards and Earth System Sciences, 2018_

## Referee Comment (RC1) · Anonymous Referee #1 · 31 Aug 2018

The paper presents a customized method to evaluate and classify ski-runs in the areas of two Canadian heli-ski companies. The problem that heli-ski companies face is that they need to reduce the risk of an accident to the maximum and at the same time minimize the limitations for their guests' experience. Therefore a sophisticated and detailed run classification can be of great value to them at the planning stage. While ATES originally targeted the "unaware" users (beginners) in their planning, the presented classification targets professional heli-ski enterprises and heli-ski guides in their specific environment, which is new. The authors use sufficiently large datasets from several

winters consisting of operationally rated runs from ski-run lists of these two companies. Furthermore, assessments of experienced heli-ski guides are used. They include terrain aspects, in respect to diverse hazards, including avalanche danger. They also use quality-aspects of guest-skiing, ski-guiding, heli-accessibility etc. Therefore, the term "terrain classification" can be misleading, use perhaps "ski-run classification for mechanized skiing operations", and instead of "terrain hierarchy" rather "ski-run hierarchy". Boot-stamped runs and very heavily tracked runs might distort the classification significantly and should be in a separate class altogether. More information on how the "runs" are defined in space (point, line, area/slope) as well as the "paths" of avalanches should be given. The method offers more detailed ratings, however the nature of the expert assessments makes it subjective and not directly transferrable to other operations. You might want to explain more clearly, how other heli-ski operations could apply the method for their settings and what the limitations are. The study's declared aim is to provide a basis for risk management. However, it is not shown, if in fact the new method could affect the risk management. Including a risk analysis (with presented accident data) would greatly enhance the paper and make it more valuable for the journal's audience. Otherwise, the risk management aspect should not be part of the paper-title nor of the aim of the study. Suggested title without risk analysis: "Deriving customized ski-run classes for two mechanized skiing operations in Canada from operational assessments" Generally, it is important to stress very strongly that a run classification is a planning tool and does not replace the ongoing re-assessment by the users in the terrain.

The paper is generally well structured, referenced and written. The two figures are difficult to follow/read in detail because very condensed. Add legend/explanation of the colour-coding in figures 1b and 2b in the figure captions. Change typing on page 10, line 2: six group -> six groups

---

## Referee Comment (RC2) · Anonymous Referee #2 · 4 Sep 2018

The paper presents different analyses made on the data collected in two mechanized skiing operations in Canada, in order" to introduce an alternative method for deriving terrain classes that offer more meaningful insight into terrain decisions in commercial mechanized skiing operations", as written at page 4 as objective of the study.

The paper is well written and clear; it tells a nice and interesting story, which the reader easily follows. In some sections it seems a bit long, but the clarity of the writing makes this not a problem. Though, this story is very much linked to the two case studies where the data were collected; the results are site-specific and cannot easily be transferred

to other places. I would discuss this aspect a bit more, not giving the impression of being too ambitious. Actually, in the discussion this limitations are well presented. . . I would then simply tell, already in the aim, that the objective is to analyse the data from the two sites in order to check if there exist possible relationships between the ski-runs considering all the characteristics listed in table 1. Also the title is ambitious. . . already there I would write something which tells the readers that this paper is based on specific case studies and does not aim at general conclusions.

Though it is case-specific, the paper is interesting as, on the contrary of other approaches (ATES, PRA identification), it used also data – explicitly said – coming from the experiences of expert guides. The used dataset, to my knowledge, is unique and deserve attention. It would be interesting to know how the guides evaluated the results and how the hierarchies will be eventually used in the future in the two mechanized skiing operations; as the paper produced practical outputs, these would be valuable. What would be interesting to check in the future is what is written at page 22 (lines 12-17): "what type of terrain is acceptable under different avalanche hazard conditions"? Maybe this concept might be expanded a bit.

Last, I am not an expert in the statistics used in the paper, therefore I would suggest to send the paper also to a statistician, who for sure will give a detailed review on the statistical methods.

---

## Author Comment (AC1) · 8 Nov 2018

**Deriving customized terrain classes for avalanche risk management in mechanized skiing operations from operational terrain assessments**

**Response to reviewers**

Reto Sterchi, Pascal Haegeli

November 07, 2018

During the review period, our manuscript received the following two anonymous referee comments:

- RC1: *'Deriving customized terrain classes for avalanche risk management in mechanized skiing operations from operational terrain assessments'*, Anonymous Referee #1, 31 Aug 2018
- RC2: *'REVIEW of manuscript by Sterchi and Haegeli'*, Anonymous Referee #2, 04 Sep 2018

We would like to thank the reviewers for taking the time to read our manuscript in detail and provide constructive feedback. The following sections describe our response to the issues raised by the two referees and outline the changes we made to the manuscript to address their concerns.

**1  Response to referee comment 1**

**1.1  Type of classification system**

**Review**
*[…] The authors use sufficiently large datasets from several winters consisting of operationally rated runs from ski-run lists of these two companies. Furthermore, assessments of experienced heli-ski guides are used. They include terrain aspects, in respect to diverse hazards, including avalanche danger. They also use quality-aspects of guest-skiing, ski-guiding, heli-accessibility etc. Therefore, the term "terrain classification" can be misleading, use perhaps "ski-run classification for mechanized skiing operations", and instead of "terrain hierarchy" rather "ski-run hierarchy". […]*

**Response to the review**
Thank you for highlighting this conceptual inconsistency. While the static, physical characteristics of the landscape describe the terrain, the other operational aspects (access, skiing experience, etc.) only come into play when we look at the terrain as ski runs that fulfill a specific operational need and are managed in a specific way. Since we are looking at avalanche terrain in a very specific context and the results are not generalizable to other activities in avalanche terrain, we agree with the suggested change in terminology. We also believe that this change might help clarify our approach for analyzing risk management decisions in a helicopter skiing operation (see comment 1.5 and 2.1).

**Changes made to the manuscript**
To address the reviewer's concern, we made the following changes (highlighted in green) throughout the manuscript.

- Replaced "terrain hierarchy" with "ski run hierarchy"
- Replaced "terrain classes" with "ski run classes"
- Adapted title: 'A method of deriving operation-specific ski run classes for avalanche risk management decisions in mechanized skiing.'

**1.2 Influence of risk mitigation activities on classification**

**Review**
*[…] Boot-stamped runs and very heavily tracked runs might distort the classification significantly and should be in a separate class altogether. […]*

**Response to the review**
We agree with the reviewer that the frequency of skiing strongly influences how runs are coded. Runs frequently coded green (i.e., open) might be assessed like this either because the terrain is relatively benign or the potential for avalanche hazard is reduced due to frequent skiing (boot-packing is a management strategy in ski resorts and is not applied in heli-skiing). We elaborate on this influential factor in the discussion of our results (see below) where we highlight that most frequently open group at CMHGL contains ski runs that are actively maintained by the guiding team to destroy potential weak layers.

> Page 19, Line 19ff (original manuscript), Section 4 Discussion
> *[…] While the physical terrain characteristics of these runs would not necessarily suggest that they belong into the group of runs that are open most often, the reason for their classification is the fact that they are actively maintained by the guiding team. Guides intentionally choose to ski these runs on a regular basis to destroy any potential weak layers before they are buried and become a risk management problem (R. Atkins, personal communication). This risk management practice allows CMHGL to have these runs open more often than their physical terrain characteristics would suggest and ski steeper terrain than on unmanaged ski runs under similar hazard conditions. These observations clearly demonstrate the ability of our approach to capture the nuanced terrain selection and risk management expertise of guides and turn them into insightful ski run hierarchies within local contexts. […]*

Please note that the focus of our approach is the identification of ski runs that are similar based on the way the are coded in the daily run lists (i.e., revealed terrain preferences reflected in daily run list ratings) and then characterize the identified groups afterwards in an independent way. The discussion of the identified groups with senior lead guides at an operation can identify such specialties.

**Changes made to the manuscript**
We hope that the issue raised in this comment is addressed by the changes made in response to some of the other comments (1.1, 1.5 and 2.1). Making the objective of our study clearer and changing the title to incorporate 'ski runs' aims to more clearly highlight that our approach focuses more comprehensively on patterns in how ski runs are opened/closed than their physical terrain characteristics alone.

**1.3 Additional background information on ski runs**

**Review**

*[…] More information on how the "runs" are defined in space (point, line, area/slope) […]*

**Response to the review**

The way runs are defined (incl. their spatial scale) varies from operation to operation. However, the common feature is that they are treated as a unit when opening or closing them in response to the expected hazard conditions during the guides meeting in the morning. We amended this detail in the introduction of our manuscript as outlined below.

**Changes made to the manuscript**

To address the reviewer's concern, we made the following changes (highlighted in green) to the manuscript.

Page 2, Line 12ff, Section 1 Introduction

*[…] During their meeting, guiding teams go through their inventory of predefined ski runs and collectively decide which runs are open or closed for skiing with guests under the expected avalanche hazard conditions. It is important to note that the scale and spatial delineation of ski runs can vary considerably from operation to operation, and there may be multiple distinct ways of skiing a run. However, ski runs are the decision units at this stage of the risk management process. […]*

**1.4 Additional background information on avalanche paths**

**Review**

*[…] as well as the "paths" of avalanches should be given. […]*

**Response to the review**

The objective of our run characterization was to collect information on key characteristics that affect guiding teams to either open or close ski runs and can help to explain the run clusters that emerged from our analysis. While a more precise delineation of start zones and avalanche paths will be critical to better understand what ski lines are chosen within runs under different types of hazard conditions, we believe that the more general, qualitative perspective focusing on whether avalanche paths are or aren't a main feature of the run in question, and at what avalanche size the ski run in question and/or the associated pickup locations become threatened seems sufficient for the objective of our study.

**Changes made to the manuscript**

To address the reviewer's concern, we made the following changes (highlighted in green) to the manuscript.

Page 8, Line 27ff, Section 2.3 Characterization of the identified run groups

*[…] We deliberately chose to mainly focus on guides' comprehensive assessment of the terrain instead of the elementary terrain parameters typically included in avalanche terrain studies. For example, instead of focusing on incline in degrees (e.g., Thumlert & Haegeli, 2018) or the precise location of exposure to avalanche paths like traditional terrain studies, our approach captures the general steepness of the run (e.g., gentle, moderately steep, moderately steep with pitches, sustained steep) and its exposure to overhead hazard (e.g., threatened during regular avalanche*

cycles, threatened during large avalanche cycles only) in a more general and qualitative perspective. This approach also allows us to gather information on more intangible ski runs characteristics that go beyond pure terrain characteristics, such as the quality of the skiing experience and the guide-ability of a run. While these guides' perspectives are associated with a certain level of subjectivity, they offer a much richer and more encompassing viewpoint of the relevant standout terrain features of ski runs that ultimately drive guiding decisions. McClung (2002) highlights the importance of human perception as a critical link or filter between observations and avalanche hazard assessment.

**1.5 Transferability of results**

**Review**
*[…] The method offers more detailed ratings, however the nature of the expert assessments makes it subjective and not directly transferable to other operations. You might want to explain more clearly, how other heli-ski operations could apply the method for their settings and what the limitations are. […]*

**Response to the review**
This comment is related to an issue raised by the second reviewer (see 2.1) and we acknowledge that we have not been clear enough in our original manuscript in addressing the transferability of our results. We think that the changes made (new title, revised statement of our objective and re-iterating our focus on the method in the conclusions, see below for details) better highlight our focus on presenting a new method to derive customized hierarchies of ski runs that can aid risk management decision-making. While run hierarchies might differ between operations (as highlighted by the two case studies included in the study), the conceptual approach for identifying groups of similarly managed runs is transferable to any other operation that works with daily run lists.

**Changes made to the manuscript**
To address the reviewer's concern, we made the following changes (highlighted in green) to the manuscript.

Title of manuscript
A method of deriving operation-specific ski run classes for avalanche risk management decisions in mechanized skiing.

Page 4, Line 16ff (original manuscript) – Section 1 Introduction
*[…] The objective of our study is to introduce an alternative and transferable method for deriving ski run classes that offer meaningful insight into terrain decisions in commercial mechanized skiing operations. Instead of building the classification from physical terrain characteristics, we derive the ski run classes from patterns in revealed terrain preferences reflected in past daily run list ratings. Our assumption is that ski runs that are considered open and closed for guiding under similar conditions will represent groupings that more closely relate to operational decision-making. We hypothesize that each operation has a unique, finely differentiated hierarchy within its ski runs that emerges from the available skiing terrain, the local snow climate and the particular skiing product it offers to its clients. […]*

Page 22, Line 15ff (original manuscript) – Section 5 Conclusions
*[…] This type of analysis has the potential to offer useful insight into what type of terrain is*

*acceptable under different avalanche hazard conditions and provide the foundation for developing evidence-based decision aids that can offer meaningful insights for terrain decisions at the ski run scale before going out into the field. […]*

Page 22, Line 12ff (original manuscript), Section 5 Conclusions

*[…]* The results of our study offer numerous contributions for future backcountry avalanche risk management research and development projects. Since a meaningful representation of terrain is critical for properly linking backcountry terrain decisions to avalanche hazard and weather conditions, the operation-specific ski run classes identified in our study provide an exciting opportunity for exploring this link. Our method of identifying ski run classes aims to overcome some of the challenges that have prevented the adoption of terrain classification systems in mechanized skiing operations in the past. While the categories of existing avalanche terrain classification system have been too broad and generic for providing meaningful assistance to professional guides, our method of identifying ski run classes aims to overcome these challenges by identifying a larger number of operation-specific terrain classes organized in a ski run hierarchies that offers a much more nuanced and applied perspective of the terrain. Even though some of the identified ski run classes might need to be further split to properly account for special risk mitigation practices (e.g., deliberate frequent skiing to manage formation of persistent weak layers), correlating avalanche conditions to the identified ski run classes has the potential to offer useful insight for the development of evidence-based decision aids that can assist guiding teams during their morning meetings. Since the patterns identified by our analysis reflect actual risk management practicies that have been used at participating operations for many years, the ski run hierarchies developed through our approach are more closely linked to the risk management decisions that the classification aims to support than existing terrain classification systems. Furthermore, the reflective nature of our approach and the fact that the emerging classification is grounded in past local risk management decisions has the potential to increase guides' acceptance and trust in the developed risk management decision aids. *[…]*

**1.6   Avalanche risk management link**

**Review**
*[…] The study's declared aim is to provide a basis for risk management. However, it is not shown, if in fact the new method could affect the risk management. Including a risk analysis (with presented accident data) would greatly enhance the paper and make it more valuable for the journal's audience. Otherwise, the risk management aspect should not be part of the paper-title nor of the aim of the study. Suggested title without risk analysis: "Deriving customized ski-run classes for two mechanized skiing operations in Canada from operational assessments". […]*

**Response to the review**
Our study has been conducted in the context of avalanche risk management by heli-skiing operations. The run list data we used to identify similarly managed ski runs primarily reflect avalanche risk management decisions. Moreover, the ultimate goal of such a classification is to enable the development of meaningful decision aids that allow operations to manage avalanche risk more efficiently (this also relates to comment 1.4 and 2.3). We therefore believe that our references to risk management are justified.

Potentially, the reviewer intended to say that the new method could affect risk instead of risk management (i.e., *The study's declared aim is to provide a basis for risk REDUCTION *). As stated by the reviewer, showing the effectiveness of the new ski run classification to reduce risk would have to be done by examining accident data before and after the implementation of the new classification system. An analysis like this is beyond the scope of this study since the ski run classifications have not been used operationally yet.

**Changes made to the manuscript**
To address the reviewer's concern, we made the following changes (highlighted in green) to the manuscript.

> Page 2, Line 10f (original manuscript), Section 1 Introduction
> *[…] In Canada, mechanized skiing operations select terrain for skiing by following a well-established process. This risk management process is iterative in nature and has been described as a series of filters occurring at multiple spatial and temporal scales (Israelson, 2013, 2015) that progressively eliminate skiing terrain from consideration. […]*

> Page 4, Line 16f (original manuscript), Section 1 Introduction
> *[…] The objective of our study is to introduce an alternative method for deriving terrain classes that offer more meaningful insight into risk management decisions in commercial mechanized skiing operations. […]*

**1.7    Run classification as planning tool**

**Review**
*[…] Generally, it is important to stress very strongly that a run classification is a planning tool and does not replace the ongoing re-assessment by the users in the terrain. […]*

**Response to the review**
It is correct that run classification system mainly assist during the morning meeting when guides are planning the skiing program for the day. This is similar to the use of existing terrain classification (e.g., Avalanche Terrain Exposure Scale) in recreational backcountry travel. During the morning meeting, the ski run classification can inform expectations and streamline the discussion of the guides when coding the runs. However, this only represents the first filter of risk management process. During the daily operations, terrain choices are further refined on an increasingly smaller scale from the run choice down to each terrain feature on a run.

**Changes made to the manuscript**
To address the reviewer's concern, we made the following changes (highlighted in green) to the manuscript.

> Page 2, Line 10f (original manuscript), Section 1 Introduction
> *[…] The resulting large-scale, consensus-based "run list" has established itself as a critical component in the risk management process of many commercial backcountry skiing operations (Israelson, 2013) and is considered best practice within the industry. The run list is a critical planning tool as it sets the stage for the skiing program of the day by eliminating certain runs from consideration. Over the course of a skiing day, terrain choices are further refined and adapted in response to direct field observations. […]*

**1.8   Clarity of figures**

**Review**
*[…] The paper is generally well structured, referenced and written. The two figures are difficult to follow/read in detail because very condensed. Add legend/explanation of the colour-coding in figures 1b and 2b in the figure captions. […]*

**Response to the review**
We acknowledge that the Figure 1 and Figure 2 are somewhat challenge to follow since they are quite condensed. To make this easier for the reader, we amended the captures of these two figures by adding the description from within the text. Having the description of how the figure is built directly underneath the figure will hopefully provide the assistance necessary to allow readers to understand the chart more easily. After careful consideration, we have come to the conclusion that adding a legend would not necessarily make the figures easier to understand. However, we would be happy to reconsider if the reviewer feels strongly about this.

**Changes made to the manuscript**
To address the reviewer's concern, we made the following changes (highlighted in green) to the manuscript.

> Caption to Figure 1 and similarly to Figure 2, Page 12 (original manuscript)
> *Figure 1: Identified terrain hierarchy with groups of similarly managed ski runs at NEH with (a) typical time series of run list ratings for the winter seasons 2013 to 2017 and (b) inter-seasonal variation within the terrain hierarchy. The time series strips of each group consist of colour-coded rows representing the run list ratings of the individual runs included in that group. Taller strips therefore represent groups with larger numbers of runs. Days when ski runs were open are shown in green, days when they were closed due to avalanche hazard are shown in red, and days when they were not discussed or closed due to non-avalanche hazard related reasons are shown in black. Days with no run list data at all (e.g., prior to operating season, days when operation was shut down due to inclement weather conditions) are shown in grey.*

**1.9   Typo**

**Review**
*[…] Change typing on page 10, line 2: six group -> six groups. […]*

**Response to the review**
The typo on page 10 has been fixed.

**Changes made to the manuscript**
To address the reviewer's concern, we made the following changes (highlighted in green) to the manuscript.

> *Page 10, Line 2 (original manuscript):*
> *[…] … we finally chose a solution with six groups of ski runs. […]*

**2 Response to referee comment 2**

**2.1 Transferability of results**

**Review**

*[…] Though, this story is very much linked to the two case studies where the data were collected; the results are site-specific and cannot easily be transferred to other places. I would discuss this aspect a bit more, not giving the impression of being too ambitious. Actually, in the discussion this limitation is well presented… I would then simply tell, already in the aim, that the objective is to analyse the data from the two sites in order to check if there exist possible relationships between the ski-runs considering all the characteristics listed in table 1. […]*

**Response to the review**

This comment and similar comments from Reviewer 1 (see Reviewer Comments 1.1 & 1.5) clearly highlight that the objective and purpose of our study was not well presented in our original submission. We hope that the revised version of our manuscript better highlights the value of our contribution.

**Changes made to the manuscript**

Please refer to our response to Reviewer Comments 1.1 and 1.5 for the changes made in the title, a clearer statement of our objective in the introduction and re-iterating our methodological focus in the conclusions.

**2.2 Title of manuscript**

**Review**

*[…] Also the title is ambitious… already there I would write something which tells the readers that this paper is based on specific case studies and does not aim at general conclusions. […]*

**Response to the review**

Similar to the previous comment of this reviewer, this comment shows that objective of our study was not clearly conveyed. In response, we changed the title of the manuscript to better highlight that our study focuses on a new method to derive customized ski run classes at mechanized skiing operations.

**Changes made to the manuscript**

To address the reviewer's concern, we made the following changes (highlighted in green) to the manuscript.

> Title of manuscript
> *A method of deriving operation-specific ski run classes for avalanche risk management decisions in mechanized skiing.*

**2.3 Application of results**

**Review**

*[…] Though it is case-specific, the paper is interesting as, on the contrary of other approaches (ATES, PRA identification), it used also data – explicitly said – coming from the experiences of expert guides. The used dataset, to my knowledge, is unique and deserve attention. It would be interesting to know how the guides evaluated the results and how the hierarchies will be eventually used in the future in the two*

*mechanized skiing operations; as the paper produced practical outputs, these would be valuable. What would be interesting to check in the future is what is written at page 22 (lines 12-17): "what type of terrain is acceptable under different avalanche hazard conditions"? Maybe this concept might be expanded a bit. […]*

**Response to the review**
The identified terrain hierarchies were well received at both operations. The general feedback was that the identified groups make sense and reflect past practices well. While the identified terrain hierarchies have not been integrated into an actual decision tool (this is the focus of our next research project), they have been used as a foundation for reflecting on patterns in terrain choices during guides trainings at the start of the season.

To address the main concern of the reviewer ([…] *It would be interesting to know how the guides evaluated the results and how the hierarchies will be eventually used* […]), we added more context about the potential application of the ski run classes in the introduction. We also expanded our description of the potential application in the conclusion section. We hope that this will make it easier for the reader to see the operational value of our study.

**Changes made to the manuscript**
To address the reviewer's concern, we made the following changes (highlighted in green) to the manuscript.

> Page 3, Line 3ff (original manuscript), Section 1 Introduction
> *[…] The vision was that the classification system would simplify the complexity of the terrain and allow guides to make appropriate terrain choices more easily. However, despite considerable efforts by CMH, the terrain classification system did not establish itself as an operational tool for making run lists. […]*

> Page 22, Line 12ff (original manuscript), Section 5 Conclusions
> *[…]* The results of our study offer numerous contributions for future backcountry avalanche risk management research and development projects. Since a meaningful representation of terrain is critical for properly linking backcountry terrain decisions to avalanche hazard and weather conditions, the operation-specific ski run classes identified in our study provide an exciting opportunity for exploring this link. Our method of identifying ski run classes aims to overcome some of the challenges that have prevented the adoption of terrain classification systems in mechanized skiing operations in the past. While the categories of existing avalanche terrain classification system have been too broad and generic for providing meaningful assistance to professional guides, our method of identifying ski run classes aims to overcome these challenges by identifying a larger number of operation-specific terrain classes organized in a ski run hierarchies that offers a much more nuanced and applied perspective of the terrain. Even though some of the identified ski run classes might need to be further split to properly account for special risk mitigation practices (e.g., deliberate frequent skiing to manage formation of persistent weak layers), correlating avalanche conditions to the identified ski run classes has the potential to offer useful insight for the development of evidence-based decision aids that can assist guiding teams during their morning meetings. Since the patterns identified by our analysis reflect actual risk management practicies that have been used at participating operations for many years, the ski run hierarchies developed through our approach are more closely linked to

the risk management decisions that the classification aims to support than existing terrain classification systems. Furthermore, the reflective nature of our approach and the fact that the emerging classification is grounded in past local risk management decisions has the potential to increase guides' acceptance and trust in the developed risk management decision aids. *[…]*